# Association of handgrip strength asymmetry and weakness with successful aging among older adults in China

Wei Ji[1]☯, Yanping Wang[2]☯, Chunping Ni[3], Xueyan Huang[4], Wenjuan He☯[5]*

1 Department of Geriatrics, XiJing Hospital, Air Force Medical University of the People's Liberation Army of China, Xi'an, Shaanxi, China, 2 Department of Medical Rehabilitation, the Third Affiliated Hospital, Air Force Medical University of the People's Liberation Army of China, Xi'an, Shaanxi, China, 3 Basic Nursing Teaching and Research Section, School of Nursing, Air Force Medical University of the People's Liberation Army of China, Xi'an, Shaanxi, China, 4 School of Nursing, Health Science Center, Xi'an Jiaotong University, Xi'an, Shaanxi, China, 5 Department of nursing, Xi'an Fifth Hospital, Xi'an, Shaanxi, China

☯ These authors contributed equally to this work.
* 15229284657@163.com

## Abstract

### Background

Successful aging (SA) is important for the increasing population aging. The role of handgrip strength(HGS) asymmetry and weakness in successful aging requires further clarification. This study aimed to elucidate the association of HGS asymmetry and weakness with successful aging in older adults.

### Methods

We included participants aged ≥60 years from the 2015 China Health and Retirement Longitudinal Study (CHARLS).SA absence of major diseases, absence of major chronic diseases, no impairment in physical function, high cognitive functioning, good mental health, and active participation in life. HGS asymmetry and weakness were measured using the maximum value of the HGS. Logistic regression modeling was used to examine the association of individuals with HGS asymmetry and weakness with SA. Restricted cubic spline (RCS) modeling was used to explore potential non-linear relationships.

### Results

Of the 5,031 individuals included, the median age of the study population was 67 years IQR: 63−73 years, 45.6% female. Only 6.3% met the criteria for successful aging. HGS asymmetry (OR = 0.597,95% CI: 0.472–0.754) and weakness (OR = 0.643,95% CI: 0.417–0.964) were both independent influences on SA. Participants were less likely to have SA when both HGD asymmetry and weakness were present (OR = 0.426,95% CI: 0.240–0.757). Further subgroup analyses revealed significant

**Data availability statement:** The data for this article comes from the the China Health and Retirement Longitudinal Study database for 2015. Available from https://charls.pku.edu.cn/en.

**Funding:** This study was supported by grants awarded to CN from the Research Project of "Clinical Medicine +X" Research Center, Air Force Medical University (LHJJ24HL02) Fund Title: Research on the Pathway of Health Behavior Empowerment and Key Nursing Technology for Elderly Patients with Chronic Diseases https://www.fmmu.edu.cn/.

**Competing interests:** The authors have declared that no competing interests exist.

associations between HGS status and each of the components of SA, particularly with regard to physical functioning. There was an inverse U-shaped relationship between HGS asymmetry and SA.

## Conclusion

HGS asymmetry is associated with a reduced likelihood of weak SA. Improving or maintaining HGS symmetry and weakness may contribute to SA in older adults.

---

## Introduction

The rising global trend of population aging has elevated successful aging (SA) as a key research focus in public health and social sciences [1]. SA encompasses the activities and behaviors through which older adults achieve positive aging outcomes despite the inevitable health declines and losses associated with aging [2]. A recent systematic review and meta-analysis estimated the global SA rate among individuals aged ≥60 years to be approximately 22% [3]. SA extends beyond merely preserving physical health; it equally emphasizes enhancing older adults' quality of life and maintaining their psychological and social well-being throughout the aging process [4]. Historically, SA research prioritized health status, particularly freedom from disease and sustained physical and mental function [2]. However, recent studies increasingly recognize non-biomedical factors as vital components of SA, including psychological resources [4], cognitive-emotional factors [5], and social engagement [6].

Within the assessment of successful aging (SA), physical activity level—a key domain [3]—serves as a widely recognized composite indicator of health and quality of life among older adults. A meta-analysis of the six most common aspects within SA criteria revealed the highest rate for absence of disability [3]. Physical activity level correlates not only with energy expenditure and metabolism [7], but also significantly with muscular endurance [8], nutritional status [9], among other factors. Handgrip Strength (HGS) is extensively utilized in aging research as a vital measure for assessing muscular endurance and nutritional status [10]. Research indicates that low HGS values show significant associations with weakness [11], functional impairment [12], malnutrition [13], and the onset of various diseases [14], underscoring the critical role of grip strength in achieving successful aging.

However, prior research has predominantly concentrated on absolute grip strength (maximum force), with less emphasis on the emerging metric of grip asymmetry—defined as a significant discrepancy between dominant and non-dominant hand strength. As a key indicator of muscle function and physical health, grip asymmetry may reflect age-related neuromuscular alterations and be linked to problems like cognitive decline [15] and functional deficits [16,17].

This study utilized nationally representative data from the China Health and Retirement Longitudinal Study (CHARLS) to assess both the independent and combined associations of HGS asymmetry and weakness with SA among Chinese adults aged

≥60 years. Associations between HGS status and individual SA domains were also explored to identify potential intervention targets.

## Methods

### Study population

The source of data for this paper is the China Health and Retirement Longitudinal Study (CHARLS) Wave 3 of 2015, which provides the most comprehensive information on successful aging in recent years. CHARLS Wave3 was approved by the Biomedical Ethics Committee of Peking University (IRB00001052–11015) and the survey was conducted with the informed consent of all participants.

The CHARLS study utilizes a multi-stage sampling approach at the county, village, household, and individual levels, with probabilities proportional to size (PPS) at each stage to ensure unbiased and representative samples. CHARLS 2015 Wave 3 included 21,095 participants, and data were screened after identifying study variables, with exclusion criteria of (1) incomplete data related to the primary study variable SA, (2) less than 60 years of age, and (3) missing data related to grip strength. A total of 16,064 participants were excluded with missing data in general information and study variables. The included data were collated and analyzed, and a total of 5,031 participants were finally included (Fig 1).

### Successful aging

According to Rowe and Kahn's definition of successful aging [2], the criteria for successful aging include the following five dimensions: absence of major chronic diseases, no impairment in physical function, high cognitive functioning, good mental health, and active participation in life. Thus, participants who met all of these conditions, including the five components listed above, were categorized as being in the Successful Aging group. Each dimension's score is set to either 0 or 1, where 0 indicates absence and 1 indicates presence. Successful aging is defined when participants have no major chronic diseases, no impairment in physical function, high cognitive function, good mental health, and active social participation in life. The single indicator of SA was operationalized as follows:

Absence of major chronic diseases: using the following series of questions: "Have you been diagnosed by a physician with any of the following medical conditions? These conditions include cancer, chronic lung disease, diabetes, heart disease, and stroke. Research has shown that the above diseases are responsible for a major burden of disease among older adults, and respondents were categorized as having no major disease if they reported no five chronic conditions [18].

No impairment in physical function: physical function was assessed using physiologically-based activities of daily living (ADL) scale. Respondents were classified as having no disability if they did not report difficulty performing any of the following six ADLs: bathing, dressing, eating, indoor transfers, toileting, or controlling urination and defecation.

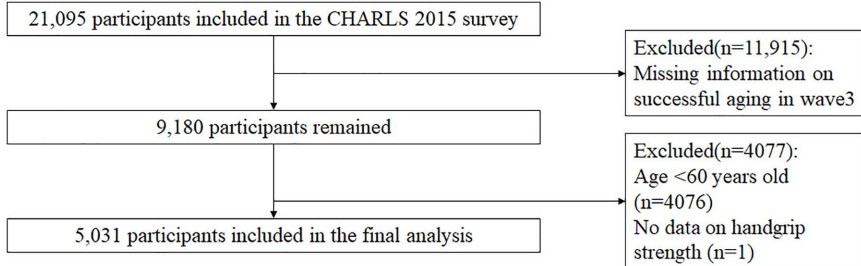

**Fig 1. Flowchart of the study.** CHARLS: China Health and Retirement Longitudinal Study; HGS: handgrip strength. wave 3:the CHARLS 2015 survey.

High Cognitive Functioning: Participants are considered to have high cognitive functioning if they achieve median or higher scores using the Telephone Interview for Cognitive Status (TICS-10), Word Recall, and Picture Drawing. The TICS-10 consists of subtracting sequences of 7 out of 100 (up to 5 times) and correctly naming the day of the week, the month, the year, and the seasons. Word recall includes immediate and delayed recall of 10 words in a list.

Good mental health: depressive symptoms are assessed using the ten items of the Center for Epidemiologic Studies Depression Scale (CESD-10). A threshold score of ≥10 was used to identify respondents with significant depressive symptoms. The absence of depression indicates good mental health, while the presence of depression indicates poor mental health.

Active social participation in life: respondents were defined as active if they participated in any of the following types of social activities: volunteer or charitable work, providing assistance to family, friends, or neighbors, or participating in sports, social, or other types of clubs in the month before the interview. This is a binary categorical variable. Individuals are defined as 'Active social participation in life' if they have engaged in any of the aforementioned social activities at least once during the past month.

### Handgrip strength asymmetry and weakness

Handgrip strength (HGS) was measured using a Yuejian™ WL-1000 mechanical dynamometer (Nantong, China) as described by Zhao et al [19]. Each participant's dominant hand was identified, and HGS was measured twice for each hand. Weakness was defined as a maximal HGS value of <28 kg for men and <18 kg for women in the dominant hand [20]. The dominant hand is determined by asking the patient 'Which is your dominant hand' [21].

To assess HGS asymmetry, the HGS ratio was calculated by dividing the maximal HGS of the non-dominant hand by that of the dominant hand. Following the "10% rule" proposed by Armstrong and Oldham [22], which suggests that HGS in the dominant hand is typically 10% stronger than in the non-dominant hand, asymmetry was defined as an HGS ratio either <0.9 or >1.1.

Participants were then classified based on the severity of their HGS asymmetry into the following categories: Normal (ratio 0.9–1.1), Mild asymmetry (ratio 0.8–0.9 or 1.1–1.2), Moderate asymmetry (ratio 0.7–0.8 or 1.2–1.3), and Severe asymmetry (ratio <0.7 or >1.3) [23].

To evaluate the combined effects of HGS asymmetry and weakness on SA, we refer to the grouping method of the study by Li et al [23]. participants were further divided into four groups: Normal (no HGS asymmetry or HGS weakness), Asymmetry only, Weakness only, and Both (HGS asymmetry and HGS weakness).

### Covariates

The following variables were included due to their potentially confounding effects on SA: age (continuous), sex (male/female), marital status (married/unmarried), place of residence (rural/urban), and educational level (Illiterate/primary/middle school/high school+).

### Statistical analyses

Means and SDs were used for continuous variables with normal distributions; medians and interquartile ranges (IQRs) were used for non-normal distributions. Frequencies and percentages were used for categorical variables. The rank-sum test and Pearson's chi-square test were used to compare sample characteristics between groups. Since SPSS's logistic regression does not directly provide the variance inflation factor (VIF), we need to approximate it using linear regression. Linear Log-Ratio Hypothesis: For continuous independent variables included in the model (e.g., age), we assessed their linear relationship with the log-ratio of the dependent variable using the Box-Tidwell test. Binary logistic regression models were used to examine the independent and combined associations of HGS asymmetry and weakness with both composite measures and different domains of SA. Two models were employed for each analysis: Model 1 was a crude model

 

without adjustment; and Model 2 was adjusted for age, gender, education level, marital status, and residence. Restricted cubic spline (RCS) regression models were employed to investigate the relationship between HGS asymmetry and the Possibilities for SA.

In addition, we performed subgroup analyses to assess the robustness of the results. Given that the association between HGS and health outcomes such as multimorbidity varies by gender, we conducted a gender-stratified analysis to explore the potential heterogeneous effects of gender. Two models were employed for each analysis: Model 1 was a crude model without adjustment; and Model 2 was adjusted for age, education level, marital status, and residence. All statistical analyses were performed using SPSS (version 25.0), except for the RCS regression model and forest plots, which were produced with R 4.4.3. Hypothesis testing was two-tailed, and statistical significance was set at $P < 0.05$.

## Results

### Characteristics of participants

Table 1 summarizes the characteristics of the study cohort. Among the 5,031 participants included in the study, the median age was 67 years (interquartile range [IQR]: 63–73 years), with 45.6% (n = 2,295) of participants being female. A total of 317 participants (6.3%) met the criteria for successful aging. Regarding specific indicators, 50.9% (n = 2,562) reported no major chronic diseases, 67.9% (n = 3,416) reported no physical dysfunction, 43.6% (n = 2,195) exhibited high cognitive functioning, 42.0% (n = 2,113) demonstrated good mental health, and 50.6% (n = 2,544) were actively engaged in life participation. In terms of handgrip strength (HGS) asymmetry, 2,632 individuals (52.3%) exhibited asymmetry, while

**Table 1. The characteristics of the participants.**

| Characteristics | Total (n = 5031) | Successfully Aging(n = 317) | Unsuccessfully Aging (n = 4714) | χ²/Z | P-value |
|---|---|---|---|---|---|
| Age, years | 67.00(63.00,73.00) | 65.00(62.50,71.00) | 67.00(63.00,73.00) | −4.523 | <0.001 |
| Gender, female, n (%) | 2295(45.6) | 170(53.6) | 2125(45.1) | 8.751 | 0.003 |
| Marital status, without a spouse, n (%) | 1044(20.7) | 38(12) | 1006(21.3) | 15.801 | <0.001 |
| Residence, rural, n (%) | 4183(83.1) | 232(73.2) | 3951(83.8) | 23.940 | <0.001 |
| Education, n (%) | | | | 20.715 | <0.001 |
| Illiterate | 469(9.3) | 8(2.5) | 461(9.8) | | |
| Primary | 549(10.9) | 43(13.6) | 506(10.7) | | |
| Middle school | 125(2.5) | 11(3.5) | 114(2.4) | | |
| High school+ | 3888(77.3) | 255(80.4) | 3633(77.1) | | |
| HGS asymmetry, n (%) | 2632(52.3) | 126(39.7) | 2506(53.2) | 21.421 | <0.001 |
| HGS weakness, n (%) | 651(12.9) | 29(9.1) | 622(13.2) | 4.317 | 0.038 |
| HGS asymmetry severity, n (%) | | | | 29.740 | <0.001 |
| Normal | 2399(47.7) | 191(60.3) | 2208(46.8) | | |
| Mild asymmetry | 1373(27.3) | 80(25.2) | 1293(27.4) | | |
| Moderate asymmetry | 561(11.2) | 28(8.8) | 533(11.3) | | |
| Severe asymmetry | 698(13.9) | 18(5.7) | 680(14.4) | | |
| HGS group,n (%) | | | | | |
| Normal | 2124(42.2) | 176(55.5) | 1948(41.3) | | |
| Asymmetry only | 2256(44.8) | 112(35.3) | 2144(45.5) | | |
| Weakness only | 275(5.5) | 15(4.7) | 260(5.5) | | |
| Both | 376(7.5) | 14(4.4) | 362(7.7) | | |

Data presented as median (P25, P75) or n (%). HGS: handgrip strength; P25: 25th percentile; P75: 75th percentile; Normal: neither weakness nor asymmetry; Both: weakness and asymmetry.

651 participants (12.9%) demonstrated weakness. The distribution of participants with mild, moderate, and severe HGS asymmetry was as follows: 1,373 (27.3%), 561 (11.2%), and 698 (13.9%), respectively. Additionally, the distribution of participants based on the presence of HGS asymmetry only, weakness only, and both asymmetry and weakness was as follows: 2,256 (44.8%), 275 (5.5%), and 376 (7.5%), respectively. Compared to individuals classified as unsuccessfully aging, those with successful aging (SA) were younger, had a higher proportion of females, were more highly educated, were less likely to be without a spouse, were less likely to reside in rural areas, and a lower prevalence of HGS asymmetry and HGS weakness (all P<0.05, Table 1).

The basic characteristics of the excluded patients are as follows. In terms of age, 8,640 patients are under 60 years old, while 7,157 are 60 years old or above. The age distribution is relatively balanced. Regarding gender, there are 7,609 male patients and 8,433 female patients, with a slightly higher number of females. In terms of educational attainment, there are 1,276 illiterate patients, 1,780 with a primary education, 790 with a middle – school education, and 12,183 with a high – school education or above. The group with a high – school education or above accounts for the largest proportion. Concerning marital status, 2,204 patients are single and 13,860 are married. The married population is in the overwhelming majority. Regarding the place of residence, 12,958 patients live in rural areas and 3,106 live in urban areas. The number of patients living in rural areas far exceeds that in urban areas.

## Associations of HGS status with SA

After multicollinearity testing, the VIF values for both models were <2, indicating no significant multicollinearity. All the Box-Tidwell test results were non-significant (p>0.05), indicating that the linear assumption holds. Individuals with both HGS asymmetry and weakness had the lowest incidence of SA compared to the normal group OR = 0.426 (95% *CI*: 0.240–0.757). Compared to the normal group, the incidence of SA decreased as the degree of grip asymmetry became more severe, with the severe asymmetry group being the least likely to have SA in this study, OR = 0.343 (95% *CI*: 0.209–0.563) (Table 2).

In addition, the associations between the detailed components of HGS status and SA are shown in Fig 2. The presence of HGS asymmetry or weakness was significantly associated with a reduced likelihood of being no impairment in the

**Table 2. Association between HGS status and successfully aging.**

| | Model1 | | Model2 | |
|---|---|---|---|---|
| | OR(95%CI) | *P*-value | OR(95%CI) | *P*-value |
| HGS asymmetry(Reference=No)<br>    Yes | 0.581(0.461,0.733) | **<0.001** | 0.597(0.472,0.754) | **<0.001** |
| HGS weakness(Reference=No)<br>    Yes | 0.662(0.448,0.979) | **0.039** | 0.634(0.417,0.964) | **0.033** |
| HGS asymmetry severity<br>(Reference=Normal) | | | | |
|     Mild asymmetry | 0.715(0.546,0.937) | **0.015** | 0.722(0.550,0.948) | **0.019** |
|     Moderate asymmetry | 0.607(0.404,0.913) | **0.017** | 0.656(0.435,0.989) | **0.044** |
|     Severe asymmetry | 0.306(0.187,0.500) | **<0.001** | 0.343(0.209,0.563) | **<0.001** |
| HGS group(Reference=Normal) | | | | |
|     Asymmetry only | 0.578(0.453,0.739) | **<0.001** | 0.620(0.484,0.795) | **<0.001** |
|     Weakness only | 0.639(0.371,1.099) | 0.105 | 0.657(0.373,1.159) | 0.147 |
|     Both | 0.428(0.246,0.746) | **0.003** | 0.426(0.240,0.757) | **0.004** |

HGS: handgrip strength; OR: odds ratio; CI: confidence interval. Model 1: a crude model; Model 2: adjusted for age, gender, education, marital status, and residence. Both: weakness and asymmetry.

| MODEL 1 | | OR(95%CI) | P-value | MODEL 2 | | OR(95%CI) | P-value |
|---|---|---|---|---|---|---|---|
| **Mental health** | | | | **Mental health** | | | |
| HGS asymmetry | | 1.224(1.094,1.369) | <0.001 | HGS asymmetry | | 1.181(1.053,1.323) | 0.004 |
| HGS weakness | | 1.026(0.869,1.212) | 0.760 | HGS weakness | | 1.568(1.294,1.900) | <0.001 |
| Asymmetry only | | 1.269(1.125,1.431) | <0.001 | Asymmetry only | | 1.221(1.080,1.382) | 0.001 |
| Weakness only | | 1.196(0.928,1.542) | 0.167 | Weakness only | | 1.806(1.375,2.372) | <0.001 |
| Both | | 1.137(0.910,1.420) | 0.259 | Both | | 1.668(1.310,2.124) | <0.001 |
| Mild asymmetry | | 1.033(0.902,1.182) | 0.642 | Mild asymmetry | | 1.023(0.892,1.174) | 0.741 |
| Moderate asymmetry | | 1.262(1.048,1.519) | 0.014 | Moderate asymmetry | | 1.232(1.020,1.488) | 0.030 |
| Severe asymmetry | | 1.655(1.397,1.961) | <0.001 | Severe asymmetry | | 1.592(1.339,1.894) | <0.001 |
| **Social participation** | | | | **Social participation** | | | |
| HGS asymmetry | | 0.864(0.773,0.965) | 0.010 | HGS asymmetry | | 0.855(0.765,0.957) | 0.006 |
| HGS weakness | | 0.763(0.647,0.900) | 0.001 | HGS weakness | | 0.788(0.654,0.951) | 0.013 |
| Asymmetry only | | 0.844(0.749,0.950) | 0.005 | Asymmetry only | | 0.844(0.748,0.952) | 0.005 |
| Weakness only | | 0.670(0.520,0.862) | 0.162 | Weakness only | | 0.701(0.536,0.917) | 0.002 |
| Both | | 0.721(0.579,0.899) | 0.072 | Both | | 0.748(0.597,0.947) | 0.004 |
| Mild asymmetry | | 0.915(0.801,1.045) | 0.188 | Mild asymmetry | | 0.907(0.794,1.125) | 0.154 |
| Moderate asymmetry | | 0.935(0.778,1.124) | 0.474 | Moderate asymmetry | | 0.935(0.776,1.125) | 0.475 |
| Severe asymmetry | | 0.723(0.611,0.857) | <0.001 | Severe asymmetry | | 0.728(0.613,0.865) | <0.001 |
| **Major chronic diseases** | | | | **Major chronic diseases** | | | |
| HGS asymmetry | | 0.857(0.767,0.958) | 0.006 | HGS asymmetry | | 0.852(0.762,0.953) | 0.005 |
| HGS weakness | | 0.922(0.782,1.087) | 0.333 | HGS weakness | | 0.953(0.790,1.149) | 0.612 |
| Asymmetry only | | 0.849(0.754,0.956) | 0.007 | Asymmetry only | | 0.855(0.758,0.964) | 0.007 |
| Weakness only | | 0.883(0.687,1.135) | 0.774 | Weakness only | | 0.928(0.711,1.212) | 0.331 |
| Both | | 0.822(0.660,1.024) | 0.759 | Both | | 0.855(0.676,1.082) | 0.080 |
| Mild asymmetry | | 0.927(0.812,1.059) | 0.265 | Mild asymmetry | | 0.930(0.813,1.063) | 0.288 |
| Moderate asymmetry | | 0.813(0.676,0.977) | 0.028 | Moderate asymmetry | | 0.821(0.681,0.988) | 0.037 |
| Severe asymmetry | | 0.766(0.647,0.907) | 0.002 | Severe asymmetry | | 0.769(0.648,0.913) | 0.003 |
| **Physical function** | | | | **Physical function** | | | |
| HGS asymmetry | | 0.702(0.623,0.791) | <0.001 | HGS asymmetry | | 0.710(0.629,0.801) | <0.001 |
| HGS weakness | | 0.649(0.548,0.768) | <0.001 | HGS weakness | | 0.599(0.493,0.729) | <0.001 |
| Asymmetry only | | 0.709(0.623,0.807) | <0.001 | Asymmetry only | | 0.739(0.648,0.842) | <0.001 |
| Weakness only | | 0.660(0.507,0.859) | <0.001 | Weakness only | | 0.636(0.479,0.843) | 0.002 |
| Both | | 0.470(0.375,0.588) | <0.001 | Both | | 0.448(0.351,0.572) | <0.001 |
| Mild asymmetry | | 0.893(0.772,1.032) | 0.126 | Mild asymmetry | | 0.895(0.773,1.036) | 0.137 |
| Moderate asymmetry | | 0.751(0.617,0.913) | 0.004 | Moderate asymmetry | | 0.783(0.643,0.955) | 0.015 |
| Severe asymmetry | | 0.433(0.364,0.515) | <0.001 | Severe asymmetry | | 0.455(0.382,0.542) | <0.001 |
| **Cognitive function** | | | | **Cognitive function** | | | |
| HGS asymmetry | | 0.727(0.650,0.813) | <0.001 | HGS asymmetry | | 0.751(0.667,0.846) | <0.001 |
| HGS weakness | | 0.840(0.710,0.994) | 0.042 | HGS weakness | | 0.638(0.524,0.776) | <0.001 |
| Asymmetry only | | 0.709(0.629,0.799) | <0.001 | Asymmetry only | | 0.780(0.684,0.888) | <0.001 |
| Weakness only | | 0.751(0.582,0.968) | 0.649 | Weakness only | | 0.619(0.467,0.819) | 0.027 |
| Both | | 0.673(0.538,0.842) | 0.659 | Both | | 0.532(0.414,0.684) | 0.001 |
| Mild asymmetry | | 0.870(0.762,0.995) | 0.041 | Mild asymmetry | | 0.885(0.767,1.022) | 0.096 |
| Moderate asymmetry | | 0.696(0.577,0.840) | <0.001 | Moderate asymmetry | | 0.764(0.624,0.935) | 0.009 |
| Severe asymmetry | | 0.518(0.433,0.618) | <0.001 | Severe asymmetry | | 0.596(0.492,0.722) | <0.001 |

**Fig 2. Association between HGS status and SA domains.** HGS: handgrip strength; OR: odds ratio; CI: confidence interval. Both: weakness and asymmetry. Model 1: a crude model; Model 2: adjusted for age, gender, education, marital status, and residence.

physical functioning domain compared to the other four domains, with ORs of 0.599 (95% *CI*: 0.472–0.754), and 0.634 (95% *CI*: 0.417–0.964), respectively. The group with both HGS asymmetry and weakness demonstrated the same effect, with OR = 0.448 (95% *CI*:0.351,0.572).

Restricted cubic spline (RCS) regression with multivariable adjustment was applied to examine the dose-response relationship between the HGS ratio and the likelihood of SA, with the shaded area representing the *95%CI* (Fig 3). As depicted in Fig 3, the RCS model identified an inverse U-shaped relationship. The odds of SA were highest at an HGS ratio of approximately 1.0 (perfect asymmetry) and decreased with greater asymmetry. The range of HGS ratios where the 95% CI for the OR included or was above 1.0 was 0.950 to 1.083, effectively refining the common clinical threshold for asymmetry.

### Subgroup and sensitivity analysis

The results of the subgroup and sensitivity analyses were consistent with our main finding that HGS asymmetry and weakness were SA influencing factors. Supplementary S1 Table presents the results stratified by gender. For men, after adjusting for all covariates, except for 'Weakness only', 'Both', and 'Moderate asymmetry', the rest of the results were significant with SA, and for the female group only the 'HGS asymmetry', 'Asymmetry only' and 'Severe asymmetry' results were significant.

### Discussion

In this study, we found that both HGS asymmetry and weakness were significantly associated with a reduced likelihood of SA among Chinese older adults. The lowest likelihood of SA was observed when both conditions were present, confirming

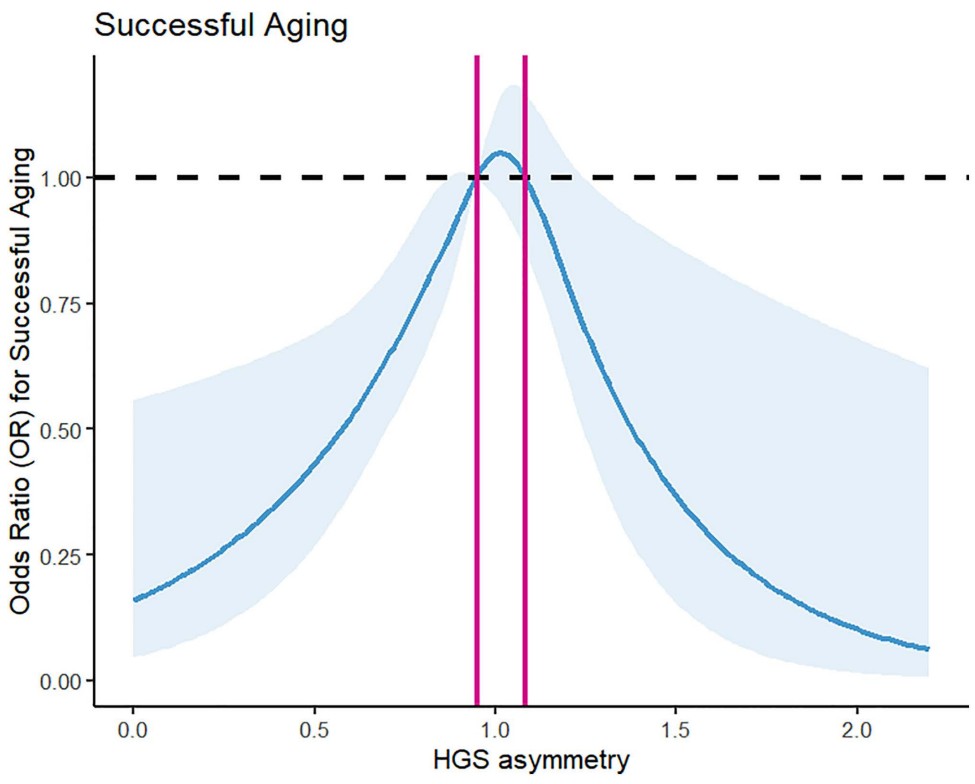

**Fig 3. Dose-response curve of HGS asymmetry and successful aging using an RCS regression model.** HGS: handgrip strength; RCS: restricted cubic spline. The two red lines indicate HGS ratios = 0.950 and 1.083, respectively, and the HGS ratio between the red lines represents a successful aging OR greater than 1. The above dose-response relationship was adjusted for following covariates: age, gender, education, marital status, and residence.

Hypothesis 1. Additionally, significant associations were identified between HGS status and each domain of SA, supporting Hypothesis 2. HGS weakness was not significantly associated with Major chronic diseases or Cognitive function. These findings highlight the importance of assessing both HGS asymmetry and weakness to identify older adults at lower risk of SA and related adverse health outcomes.

Aligning with prior research, HGS asymmetry or weakness shows strong links to declines in both physical and cognitive function [12,16,24,25]. At the biological level, key contributors to reduced grip strength in older adults include muscle mitochondrial dysfunction, diminished muscle protein synthesis, and decreased levels of critical hormones like testosterone and growth hormone [24,26]. Decreasing muscle mass leads to concurrent declines in grip strength, primarily affecting older individuals' physical mobility [18]. Consequently, preventive and management strategies aimed at maintaining grip strength and correcting imbalances could yield multifaceted benefits. These interventions not only enhance energy metabolism and vitality but also may positively influence other SA-related factors. A dual-strategy combining targeted exercise and specific nutrition is critical to counter sarcopenia-driven handgrip strength (HGS) decline. Exercise prescriptions should emphasize unilateral movements (e.g., dumbbell curls, single-arm rows, farmer's carries) over bilateral exercises to eliminate compensatory dominance and ensure balanced strength development. Adherence to volume-matched training—where both limbs perform the same sets and repetitions, irrespective of initial load—is fundamental. Concurrently, dietary strategies must synergize with training by providing sufficient protein to support muscle synthesis and key nutrients like Vitamin D to address the metabolic underpinnings of muscle loss. The focus should be on fatiguing each side independently to a similar degree. This is especially vital for elderly individuals with significant grip asymmetry, as the current study observed the lowest sarcopenia likelihood in those with normal grip strength compared to those with impairments.

This study confirmed strong associations between both HGS asymmetry/weakness and SA in older Chinese adults, with asymmetry being universally associated across all SA domains. First, our results corroborate prior research indicating HGS decline predicts physical disability and loss of independence [16]. Asymmetric grip has specifically been linked to motor impairments, including reduced gait speed and poorer balance [23]. It may also signify imbalanced hemispheric activation or neurological dysfunction [27]. Multiple studies associate increased HGS asymmetry with higher cognitive decline risk [20,28]. Second, HGS asymmetry exhibits a robust association with mental health, particularly depression. Existing evidence demonstrates links between both asymmetry/low HGS and depression, with the strongest association occurring when both factors coexist [23,29]. This relationship may stem from the impact of asymmetric hemispheric activation on cognitive domains like language, spatial attention, and social perception [30]. Furthermore, HGS asymmetry may negatively affect social participation. This connection could be mediated by the established link between asymmetry and diminished physical function [12], potentially leading individuals to curtail social activities. Finally, asymmetric grip may influence major disease occurrence in the elderly. Consistent with previous findings [31], both HGS weakness and asymmetry could elevate cardiovascular risk. As this study did not examine specific diseases, future research should explore potential links between grip asymmetry and particular major illnesses in older populations.

Finally, our analysis revealed the lowest SA likelihood among individuals exhibiting both HGS asymmetry and weakness, with this effect most pronounced in males during gender subgroup analysis. The occurrence of HGS asymmetry exceeded twofold that of weakness, indicating it may serve as an early indicator of muscle dysfunction preceding overall strength decline [32]. An HGS ratio within 0.950–1.083 correlated with higher SA odds (odds ratio > 1), refining the relevant asymmetry range for SA analysis.

However, several limitations warrant consideration. First, the cross-sectional nature of our study design fundamentally precludes the determination of causal relationships between handgrip strength states and successful aging. The observed associations, while robust, may reflect reverse causality or be driven by unmeasured underlying factors. Second, the exclusion of females with HGS weakness precluded observing gender differences consistent with subgroup findings. Third, similar to other research [33], major disease diagnoses relied on self-reported physician assessments. Absence of medical records in CHARLS means self-reported chronic conditions may introduce reporting bias. Fourth, as previously

discussed, it is probable that residual confounding persists. Factors such as lifelong socioeconomic status, genetic predisposition, chronic inflammation, and psychosocial stress, which were not measured in our study, may influence both musculoskeletal health and aging trajectories, confounding the observed relationships. Finally, the conceptual foundation of "successful aging" itself warrants critical reflection. The application of the Rowe and Kahn model, while enabling cross-study comparison, may not fully capture culturally specific dimensions of aging well in the Chinese context, such as familial harmony and spiritual contentment. This may affect the content validity of our outcome measure. Despite these constraints, this study provides significant insights into the understudied links between grip asymmetry, weakness, and SA in China's elderly population, advancing understanding of these factors within aging research.

## Conclusions

This study suggests that the presence of HGS asymmetry and weakness is independently associated with a reduced likelihood of SA, and that HGS asymmetry is independently associated with each component of SA, with the lowest likelihood of SA occurring when both are present. Maintaining HGS symmetry and reducing asymmetry may help promote SA in older adults.

## Supporting information

**S1 Table. Association between HGS status and SA in different gender groups.**
(DOCX)

## Acknowledgments

We thank Peking University for the open data resources and all investigators who participated in the study.

## Author contributions

**Conceptualization:** Wei Ji, Yanping Wang, Chunping Ni, Xueyan Huang, wenjuan He.

**Data curation:** Wei Ji, Yanping Wang, Chunping Ni, wenjuan He.

**Formal analysis:** Wei Ji.

**Methodology:** Wei Ji.

**Writing – original draft:** Wei Ji, Yanping Wang, wenjuan He.

**Writing – review & editing:** Wei Ji, Yanping Wang, Chunping Ni, Xueyan Huang, wenjuan He.

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
