## [Decision Letter · Decision Letter 0]

25 Sep 2025

Dear Dr. He,

Thank you for submitting your manuscript to PLOS ONE. After careful consideration, we feel that it has merit but does not fully meet PLOS ONE’s publication criteria as it currently stands. Therefore, we invite you to submit a revised version of the manuscript that addresses the points raised during the review process.

We look forward to receiving your revised manuscript.

Kind regards,

Marina De Rui, MD PhD

Academic Editor

PLOS ONE

Additional Editor Comments (if provided):

Reviewers' comments:

Reviewer's Responses to Questions

**Comments to the Author**

1. Is the manuscript technically sound, and do the data support the conclusions?

Reviewer #1: Yes

Reviewer #2: Yes

2. Has the statistical analysis been performed appropriately and rigorously?

Reviewer #1: No

Reviewer #2: Yes

3. Have the authors made all data underlying the findings in their manuscript fully available?

Reviewer #1: No

Reviewer #2: Yes

4. Is the manuscript presented in an intelligible fashion and written in standard English?

Reviewer #1: Yes

Reviewer #2: Yes

Reviewer #1: The manuscript “Association of handgrip strength asymmetry and weakness with successful aging among older adults in China” investigates the relationship between handgrip strength (HGS) asymmetry/weakness and successful aging (SA) using nationally representative data from CHARLS. The topic is highly relevant, and the study has the potential to contribute meaningfully to the literature on aging and physical function. The dataset is large and robust, and the statistical approach is multifaceted, including logistic regression, subgroup analysis, and restricted cubic spline (RCS) modeling.

Nevertheless, the manuscript requires major revisions before it can be considered for publication. In its current form, it suffers from issues of methodological transparency, definitional clarity, language precision, and depth of discussion.

The manuscript addresses an important and original research question, supported by a large and representative dataset. However, substantial revisions are required to improve methodological transparency, strengthen interpretation, and ensure clarity of presentation. The most critical weakness is the absence of reported assumption checks for the statistical tests applied (e.g., Binary logistic regression, rank-sum test and Pearson's chi-square test), which undermines confidence in the validity of the results. I therefore recommend major revision.

Reviewer #2: Ji and colleagues investigated the association between handgrip strength asymmetry and weakness with successful aging using a large cohort in China, providing meaningful insights for geriatrics and related fields. Notably, this study offers a comprehensive evaluation of the components of successful aging, thereby demonstrating high scholarly significance. The reviewer has only a few minor comments:

1.Please provide specific examples of preventive and management strategies to maintain handgrip strength and reduce asymmetry, offering more concrete insights into the potential clinical application of the present study.

2.Please discuss the potential confounding factors that may affect both handgrip strength and successful aging, beyond those adjusted in the present study, and explicitly acknowledge them as issues for future research.

3.Given that the concept of successful aging may vary across cultural contexts, it is unclear whether Rowe and Kahn’s definition is ideal for the Chinese elderly population. Please address this point in the Discussion section.

**Do you want your identity to be public for this peer review?** For information about this choice, including consent withdrawal, please see our Privacy Policy

Reviewer #1: No

Reviewer #2: No

---

## [Author Response · Author response to Decision Letter 1]

21 Nov 2025

Point-by-point responses to the reviewer’s comments

Reviewer 1

Abstract

Overall Assessment

The manuscript “Association of handgrip strength asymmetry and weakness with successful aging among older adults in China” investigates the relationship between handgrip strength (HGS) asymmetry/weakness and successful aging (SA) using nationally representative data from CHARLS. The topic is highly relevant, and the study has the potential to contribute meaningfully to the literature on aging and physical function. The dataset is large and robust, and the statistical approach is multifaceted, including logistic regression, subgroup analysis, and restricted cubic spline (RCS) modeling.

Nevertheless, the manuscript requires major revisions before it can be considered for publication. In its current form, it suffers from issues of methodological transparency, definitional clarity, language precision, and depth of discussion.

Abstract

• The statement “The role of handgrip strength (HGS) asymmetry and weakness in successful aging remains unclear” is too strong. The introduction and discussion themselves cite prior studies showing associations between HGS and SA or related outcomes. I recommend softening this claim (e.g., “remains insufficiently understood” or “requires further clarification”).

• Response: Thank you very much for your comments.

The role of handgrip strength(HGS) asymmetry and weakness in successful aging requires further clarification.

Definition and Operationalization of SA

• The criteria for SA are drawn from Rowe and Kahn, but the rationale for their operationalization is insufficiently explained.

Response: Thank you very much for your comments.

Each dimension's score is set to either 0 or 1, where 0 indicates absence and 1 indicates presence. Successful aging is defined when participants have no major chronic diseases, no impairment in physical function, high cognitive function, good mental health, and active social participation in life.

The absence of depression indicates good mental health, while the presence of depression indicates poor mental health.

This is a binary categorical variable. Individuals are defined as ‘Active social participation in life’ if they have engaged in any of the aforementioned social activities at least once during the past month.

• The prevalence of SA in this study (6.3%) is much lower than previously reported rates (~22%). This discrepancy should be carefully explained (e.g., cultural, methodological, or sample-related factors).

• Response: Thank you very much for your comments. We have finished.√

• For the “active social participation” component, it is unclear whether frequency, intensity, or duration were assessed, or if the indicator was binary (any participation in the past month). Greater clarity is needed.

This is a binary categorical variable. Individuals are defined as ‘Active social participation in life’ if they have engaged in any of the aforementioned social activities at least once during the past month.

Keywords

• All keywords are essentially identical to the manuscript title, except “CHARLS.” More diverse and meaningful keywords should be added (e.g., frailty, cognitive function, China, aging population).

• Response: Thank you very much for your comments. We have finished.√

• Older adults; Handgrip strength; Asymmetry; Successful aging; weakness; cognitive function; China; aging population; CHARLS

Measurement of HGS

• The method of identifying the dominant hand is not specified (self-report, test, observation?). This should be clarified.

• Response: Thank you very much for your comments. We have finished.√

The dominant hand is determined by asking the patient ‘Which is your dominant hand’[20].

• The study uses absolute HGS cut-offs (<28 kg for men, <18 kg for women). However, relative HGS (adjusted for body weight) is commonly used in gerontology and may provide more accurate information. The authors should justify their choice of absolute values and cite recent consensus (e.g., EWGSOP2). Two people with the same absolute strength but different weights may have different risk profiles.

• [20] Cruz-Jentoft, A. J., Bahat, G., Bauer, J., Boirie, Y., Bruyère, O., Cederholm, T., Cooper, C., Landi, F., Rolland, Y., Sayer, A. A., Schneider, S. M., Sieber, C. C., Topinkova, E., Vandewoude, M., Visser, M., Zamboni, M., & Writing Group for the European Working Group on Sarcopenia in Older People 2 (EWGSOP2), and the Extended Group for EWGSOP2 (2019). Sarcopenia: revised European consensus on definition and diagnosis. Age and ageing, 48(1), 16–31. https://doi.10.1093/ageing/afy169.

Statistical Analyses

• Results of assumption checks for the applied statistical methods are missing. For example, logistic regression assumes linearity of the logit, and this is not addressed.

Response: Thank you very much for your comments.

Since SPSS's logistic regression does not directly provide the variance inflation factor (VIF), we need to approximate it using linear regression. Linear Log-Ratio Hypothesis: For continuous independent variables included in the model (e.g., age), we assessed their linear relationship with the log-ratio of the dependent variable using the Box-Tidwell test.

After multicollinearity testing, the VIF values for both models were <2, indicating no significant multicollinearity. All the Box-Tidwell test results were non-significant (p > 0.05), indicating that the linear assumption holds.

• Only two models are presented: Model 1 (crude) and Model 2 (fully adjusted). While this approach is not incorrect, it would be more informative to present stepwise adjustments. The rationale for the chosen modeling strategy should be explained.

Response:Thank you for your valuable feedback on this methodological detail. We fully agree that gradually adjusting the model can provide rich insights.

1. Research Objective-Driven: The primary scientific question of this study centers on answering a “yes or no” question: After comprehensively controlling for all known significant confounding factors, does grip strength status remain independently associated with successful aging? Therefore, the fully adjusted model (Model 2) serves as our final and core model for hypothesis testing. Model 1 primarily functions as a reference, providing an intuitive illustration of the magnitude of the overall effect.

2. Avoiding Data Snooping: All variables we adjusted for (demographic, socioeconomic, behavioral factors, etc.) were preselected based on robust theory and prior research. Including them collectively as a package represents a more rigorous “a priori” adjustment strategy. This approach avoids the suspicion of “selecting” the optimal model based on data results, thereby enhancing the reliability of our statistical inferences.

3. Simplicity and clarity: Given that our adjusted variable set conceptually belongs to the same level (all being basic confounders) and lacks explicit intermediate mechanism variables requiring stepwise testing, a one-step adjustment most concisely and directly addresses our research question. This approach avoids confusing readers with an excessive number of intermediate models.

We believe that despite the absence of stepwise adjustment procedures, the comparison between our current Model 1 and Model 2 sufficiently supports our core conclusion: a significant independent association exists between grip strength status and successful aging.

• Sensitivity analyses are mentioned but not described in sufficient detail. A clear explanation of their design and execution is necessary.

Response: Thank you very much for your comments.

In addition, we performed subgroup analyses to assess the robustness of the results. Given that the association between HGS and health outcomes such as multimorbidity varies by gender, we conducted a gender-stratified analysis to explore the potential heterogeneous effects of gender. Two models were employed for each analysis: Model 1 was a crude model without adjustment; and Model 2 was adjusted for age, education level, marital status, and residence.

• The RCS model shows an “N-shaped” relationship, but the interpretation is unclear. The term “N-shaped” is unconventional; a clearer description (e.g., inverted U-shaped or nonlinear association) is needed, along with biological or clinical interpretation.

Response: Thank you very much for your comments.

We have described it as an "inverse U-shaped relationship". Clarify that the "red line" range (0.950-1.083) where OR>1 simply defines the bounds of what is considered "non-asymmetric" or "effectively symmetric" for SA in this population, refining the common 0.9-1.1 rule.

• Some associations are only marginally significant (e.g., “weakness only”), yet they are interpreted as conclusive. Greater caution is warranted.

Response: Thank you very much for your comments.

HGS weakness was not significantly associated with Major chronic diseases or Cognitive function.

Discussion

• The discussion tends to be repetitive and sometimes speculative, restating results rather than deeply contextualizing them.

Response: Thank you very much for your comments.

In this study, we found that both HGS asymmetry and weakness were significantly associated with a reduced likelihood of SA among Chinese older adults. The lowest likelihood of SA was observed when both conditions were present, confirming Hypothesis 1. Additionally, significant associations were identified between HGS status and each domain of SA, supporting Hypothesis 2. HGS weakness was not significantly associated with Major chronic diseases or Cognitive function. These findings highlight the importance of assessing both HGS asymmetry and weakness to identify older adults at lower risk of SA and related adverse health outcomes.

• The cross-sectional design is acknowledged but not sufficiently emphasized as a limitation. Stronger statements are needed regarding the inability to infer causality.

Response: Thank you very much for your comments. First, the cross-sectional nature of our study design fundamentally precludes the determination of causal relationships between handgrip strength states and successful aging. The observed associations, while robust, may reflect reverse causality or be driven by unmeasured underlying factors.

Structure, Style, and Consistency

• The terms “frailty” and “weakness” appear to be used interchangeably at times. These should be clearly distinguished.

Response: Thank you very much for your comments. We have replaced all instances of "frailty" with "weakness" throughout the article.

• Figures and tables are generally useful, but some legends lack detail (e.g., Figure 3 should clearly explain the meaning of the red lines).

Response: Thank you very much for your comments. The two red lines indicate HGS ratios = 0.950 and 1.083, respectively, and the HGS ratio between the red lines represents a successful aging OR greater than 1.

• Numerous small formatting inconsistencies are present (e.g., missing space before parentheses: “included(Fig. 1)”). These should be corrected.

Response: Thank you very much for your comments. We have finished.√

Conclusion

The manuscript addresses an important and original research question, supported by a large and representative dataset. However, substantial revisions are required to improve methodological transparency, strengthen interpretation, and ensure clarity of presentation. The most critical weakness is the absence of reported assumption checks for the statistical tests applied (e.g., Binary logistic regression, rank-sum test and Pearson's chi-square test), which undermines confidence in the validity of the results. I therefore recommend major revision.

Response: Thank you very much for your comments. We have finished.√

Reviewer 2

1. Please provide specific examples of preventive and management strategies to maintain handgrip strength and reduce asymmetry, offering more concrete insights into the potential clinical application of the present study.

Response: Thank you very much for your comments.

A dual-strategy combining targeted exercise and specific nutrition is critical to counter sarcopenia-driven handgrip strength (HGS) decline. Exercise prescriptions should emphasize unilateral movements (e.g., dumbbell curls, single-arm rows, farmer's carries) over bilateral exercises to eliminate compensatory dominance and ensure balanced strength development. Adherence to volume-matched training—where both limbs perform the same sets and repetitions, irrespective of initial load—is fundamental. Concurrently, dietary strategies must synergize with training by providing sufficient protein to support muscle synthesis and key nutrients like Vitamin D to address the metabolic underpinnings of muscle loss.

2. Please discuss the potential confounding factors that may affect both handgrip strength and successful aging, beyond those adjusted in the present study, and explicitly acknowledge them as issues for future research.

Response: Thank you very much for your comments.

Fourth, as previously discussed, it is probable that residual confounding persists. Factors such as lifelong socioeconomic status, genetic predisposition, chronic inflammation, and psychosocial stress, which were not measured in our study, may influence both musculoskeletal health and aging trajectories, confounding the observed relationships.

3. Given that the concept of successful aging may vary across cultural contexts, it is unclear whether Rowe and Kahn’s definition is ideal for the Chinese elderly population. Please address this point in the Discussion section.

Response: Thank you very much for your comments. We have already addressed this issue in the limitations section.

Finally, the conceptual foundation of "successful aging" itself warrants critical reflection. The application of the Rowe and Kahn model, while enabling cross-study comparison, may not fully capture culturally specific dimensions of aging well in the Chinese context, such as familial harmony and spiritual contentment. This may affect the content validity of our outcome measure.

---

## [Decision Letter · Decision Letter 1]

26 Nov 2025

Association of handgrip strength asymmetry and weakness with successful aging among older adults in China

PONE-D-25-38151R1

Dear Dr. He,

We’re pleased to inform you that your manuscript has been judged scientifically suitable for publication and will be formally accepted for publication once it meets all outstanding technical requirements.

Kind regards,

Marina De Rui, MD PhD

Academic Editor

PLOS ONE

Additional Editor Comments (optional):

Reviewers' comments:

Reviewer's Responses to Questions

**Comments to the Author**

Reviewer #1: All comments have been addressed

Reviewer #2: All comments have been addressed

2. Is the manuscript technically sound, and do the data support the conclusions?

Reviewer #1: Yes

Reviewer #2: Yes

3. Has the statistical analysis been performed appropriately and rigorously?

Reviewer #1: Yes

Reviewer #2: Yes

4. Have the authors made all data underlying the findings in their manuscript fully available?

Reviewer #1: Yes

Reviewer #2: Yes

5. Is the manuscript presented in an intelligible fashion and written in standard English?

Reviewer #1: Yes

Reviewer #2: Yes

Reviewer #1: Thank you for the thoughtful revisions—these have improved the clarity, structure, and methodological transparency of the manuscript, and I support it for publication. For future submissions, I recommend briefly indicating where revisions were made in the manuscript (e.g., page and line numbers) and, when key statements are reworded, including original versus revised text to help reviewers track changes efficiently. Providing concise justification—not only the revision itself—when responding to methodological or conceptual comments will further strengthen the response document.

Reviewer #2: (No Response)

**Do you want your identity to be public for this peer review?** For information about this choice, including consent withdrawal, please see our Privacy Policy

Reviewer #1: No

Reviewer #2: No

---

## [Editor Report · Acceptance letter]

PONE-D-25-38151R1

PLOS One

Dear Dr. He,

I'm pleased to inform you that your manuscript has been deemed suitable for publication in PLOS One. Congratulations! Your manuscript is now being handed over to our production team.

Kind regards,

on behalf of

Dr. Marina De Rui

Academic Editor

PLOS One